# Removal of Aerosolized Contaminants from Working Canines via a Field Wipe-Down Procedure

**DOI:** 10.3390/ani11010120

**Published:** 2021-01-08

**Authors:** Erin B. Perry, Dakota R. Discepolo, Stephen Y. Liang, Eileen K. Jenkins

**Affiliations:** 1Department of Animal Science Food & Nutrition, Southern Illinois University, 1205 Lincoln Drive, Carbondale, IL 62901, USA; Dakota.Discepolo@siu.edu; 2Department of Emergency Medicine, Washington University School of Medicine, St. Louis, MO 63110, USA; syliang@wustl.edu; 3Division of Infectious Diseases, Department of Medicine, Washington University School of Medicine, St. Louis, MO 63110, USA; 4VCA Regional Institute for Veterinary Emergencies and Referrals, Chattanooga, TN 37406, USA; eileen.jenkinsdvm@gmail.com

**Keywords:** working dog, decontamination, aerosolized contaminants

## Abstract

**Simple Summary:**

Aerosolized particulates are potential sources of contamination for working dogs across a variety of environments. This work describes a simple wipe-down procedure using common veterinary antiseptic cleansers that may be effective for decontamination purposes. Dilute povidone-iodine scrub wipes were more effective than dilute chlorhexidine-gluconate scrub wipes or water at removal of aerosolized water-borne particulates from the coat of working dogs. Coat differences associated with common breeds did not change efficacy of the wipe-down procedure or cleanser. These novel data describe a simple, field expedient procedure for a wipe-down that may effectively reduce aerosolized particulates on the coats of working dogs.

**Abstract:**

Evidence-based canine decontamination protocols are underrepresented in the veterinary literature. Aerosolized microbiological and chemical contaminants can pose a risk in deployment environments highlighting the need for improved canine field decontamination strategies. Prior work has established the efficacy of traditional, water-intensive methods on contaminant removal from the coat of the working canine; however, it is not known if similar reductions can be achieved with simple field expedient methods when resources are limited. The objective of this study was to measure the reduction of aerosolized contamination via a practical “wipe-down” procedure performed on working canine coats contaminated with a fluorescent, non-toxic, water-based aerosol. Disposable, lint-free towels were saturated with one of three treatments: water, 2% chlorhexidine gluconate scrub (CHX), or 7.5% povidone-iodine scrub (PVD). Both CHX and PVD were diluted at a 1:4 ratio. Treatments were randomly assigned to one of three quadrants established across the shoulders and back of commonly utilized working dog breeds (Labrador retrievers, n = 16; German shepherds, n = 16). The fourth quadrant remained unwiped, thus serving as a control. Reduction in fluorescent marker contamination was measured and compared across all quadrants. PVD demonstrated greater marker reduction compared to CHX or water in both breeds (*p* < 0.0001). Reduction was similar between CHX or water in Labradors (*p* = 0.86) and shepherds (*p* = 0.06). Effective wipe-down strategies using common veterinary cleansers should be further investigated and incorporated into decontamination practices to safeguard working canine health and prevent cross-contamination of human personnel working with these animals.

## 1. Introduction

Working canines frequently operate in contaminated environments. Toxicologic hazards for working dogs include gases (e.g., hydrogen cyanide and nitrogen dioxide), solids and/or liquids (e.g., polychlorinated biphenyls and toxic metals), and particulates (e.g., asbestos and fiberglass fibers) [1]. Contact with aerosolized contaminants is possible when operations occur near contaminated waters (e.g., floodwater, sewage). Such aerosols may harbor pathogenic microorganisms [2] and/or hazardous chemicals that can contaminate the exterior coat of the working canine, posing direct health risks to the canine through inhalation, ingestion, and skin contact. Human personnel are also at risk through fomite transmission. Health risks associated with contamination include acute injury or illness [3] and chronic illness including cancer [1]. While any canine may be at risk, breeds at greatest risk for exposure include those commonly utilized for working disciplines including Labrador retrievers and German shepherds. Evidence-based decontamination strategies are needed to mitigate the potential health risks posed by aerosolized contaminants present on the coat of the working canine.

Decontamination of working canines deployed to contaminated environments has been recommended previously and should be incorporated into training protocols and best practices [4,5,6]. Current canine decontamination procedures utilize significant amounts of water, which may not be available in resource-limited settings. This study utilizes a 0.05% solution (2% chlorhexidine surgical scrub prepared as a 1:4 dilution) based on demonstrated efficacy in prior studies. Decolonization of the canine coat in therapy dogs used for hospital visits was achieved using a combination of chlorhexidine-based shampoo and chlorhexidine wipes during a study of hospital-associated infections [7]. Povidone-iodine is a topical iodophore disinfectant that has broad spectrum biocidal activity against gram-negative and gram-positive bacteria, spores, mycoplasmas, enveloped and non-enveloped viruses, and fungi. Povidone-iodine surgical scrub is fast acting and has detergent properties [5]. This study utilizes a 1.875% solution (7.5% povidone-iodine surgical scrub prepared as 1:4 dilution; 18,750 ppm), which exceeds the minimum 50–100 parts per million (ppm) bactericidal threshold [8] but is significantly less than the 100,000 ppm (10%) threshold associated with diminished tissue healing and mammalian cellular tissue damage [9]. These simple field expedient decontamination agents on disposable towels were evaluated for efficacy in reducing the burden of a simulated aerosol contaminant on the exterior coat of the working canine. In addition to ease of use, this water-restrictive method also reduces the likelihood of wash-in effect by removing surface contamination directly and minimizes disruption of the canine’s epidermal barrier which may occur with full-body bathing and traditional water-intensive decontamination. We hypothesize that our method effectively removes a simulated water-based aerosol contaminant from the coat of working canines in resource-limited settings and may preserve dermal health.

## 2. Materials and Methods

### 2.1. Animal Enrollment

Institutional Animal Care & Use approval (#19-031) was obtained from Southern Illinois University prior to the initiation of this study. Due to their prevalence within the working dog community, we selected two common working breeds (Labrador retrievers, n = 16; and German shepherds, n = 17) from two facilities for inclusion in this study. Black Labradors (upland sporting dogs) from a single kennel, housed in similar conditions and maintained on a single commercially available balanced diet (chicken and rice) were included. German shepherds (military/law enforcement dogs) from a single facility, housed in similar conditions and maintained on a single commercially available, balanced diet (chicken and rice) were included. All canines were assessed for health by a licensed veterinarian prior to inclusion in the study. One dog was removed from the study due to the presence of a skin lesion at the anatomical site where the decontamination methods were to be evaluated. Results are presented for the remaining 32 canines.

### 2.2. Application of Contaminant and Wipe-Down Procedure

The dorsal aspect of each canine was divided into quadrants (Figure 1) and dermal pH was measured at the base of the tail in triplicate using a hand-held dermal pH meter (HI 99181 Portable Waterproof Skin pH Meter, Hanna Instruments, Woonsocket, RI, USA) to document pre-existing dermal conditions prior to application of the simulated aerosol contaminant.

A simulated, non-toxic fluorescent contaminant (GloGerm^®^, Moab, UT, USA) was combined with water to create a 1:8 ratio solution. A commercially available sprayer (Master Blaster, Bottle Crew Farmington Hills, MI, USA) was used to aerosolize and apply the simulated contaminant. The sprayer was held at a distance of 60 cm (±5) from the canine’s hips and directed parallel from the rear to the front of the canine to create an aerosolized contamination of 49 (±11) droplets/cm^2^. An Elizabethan collar was used to protect the canine’s head from exposure.

Following application of the contaminant, disposable, lint-free towels (Davelen^©^; Derwood, MD, USA) were saturated and utilized for wipe-down decontamination with one of the following three treatments: water (H_2_O), 2% chlorhexidine gluconate scrub (CHX), or 7.5% povidone-iodine scrub (PVD). The two cleansers were diluted with water at a ratio of 1:4. Wipe-down with one of the three treated towels was randomly assigned to a quadrant on the dorsal aspect of the canine. The last remaining quadrant was left unwiped to serve as a control for comparison of contaminant reduction. Following cleanser wipe-down, a second wipe was performed using a water-saturated towel to remove any antiseptic cleanser residue.

Fluorescence, indicative of simulated aerosol contamination, was documented in each of the quadrants on the dorsal aspect of the canine following wipe-down and compared with the control quadrant via digital imaging using a Canon T5i DSLR (Canon Inc., Tokyo, Japan) camera positioned 45 cm (±5) from the canine. The removal of droplets (i.e., physical reduction of simulated contaminant) was scored as fluorescence reduction utilizing a method previously published [10,11,12]. The scoring method was applied utilizing two blinded and independent reviewers with 82% agreement. Contamination reduction scores were defined as follows: 0 ≤ 25% contamination reduction; 1 = 25–50% contamination reduction; 2 = 51–75% contamination reduction; 3 ≥ 75% contamination reduction. No score discrepancies >1 were observed between reviewers. A score of 3 (±75% reduction) was considered successful decontamination. Contaminant reduction scores for each treatment are reported as a percentage of total frequency (192 total scores).

### 2.3. Statistical Analysis

Data entry was performed using Microsoft Excel (Microsoft Corporation, Redmond WA, USA) and data were analyzed using SAS, version 9.4 (SAS Institute Inc., Cary, NC, USA). Dermal pH data were analyzed using PROC GLM, and categorical data for fluorescence reduction were analyzed using PROC FREQ. Significance for all variables of interest was established at *p* < 0.05.

## 3. Results

### 3.1. Dermal pH

Dermal pH of study participants was unaffected by breed when shepherds were compared to retrievers (*p* = 0.3393). Mean dermal pH was 8.32 and 8.60 for German shepherds and Labrador retrievers, respectively. Additionally, no differences in dermal pH were observed between intact males (8.38), intact females (8.73), or spayed females (8.35) (*p* = 8.30). No effect was evident for breed*sex (*p* = 0.8175).

### 3.2. Contaminant Reduction

Contaminant reduction was similar between Labrador retrievers and German shepherds (*p* = 0.6417; Figure 2). Amongst Labrador retrievers, the greatest frequency of successful decontamination scores was observed with PVD (success = 26) compared to CHX (success = 8) or water (success = 7) wipes (*p* < 0.0001). Similarly, amongst German shepherds, the highest frequency of successful decontamination was also seen with PVD scores (success = 21) as compared to CHX (success = 3) or water (success = 6) (*p* < 0.0001, Table 1).

Frequency of scores for each cleanser indicating successful contaminant removal are shown in Figure 3. Overall, cleanser treatment significantly impacted contaminant reduction for study participants (*p* < 0.0001) with a greater rate of successful reduction (68.75%) associated with PVD treatment. Furthermore, when CHX (20% successful scores) was compared to water (23% successful scores), no significant difference in success scores was observed (*p* = 0.4568).

## 4. Discussion

Evidence-based field decontamination strategies are needed to address the wide range of environmental hazards working canines are likely to encounter during disaster, search and rescue, law enforcement, and national security responses. While inhalation of hazardous aerosols remains difficult to prevent in working canines, aerosol contamination of the exterior coat with water-based aerosols can be mitigated. In this study, a simple wipe-down procedure using disposable towels saturated with diluted 7.5% povidone-iodine scrub, a common veterinary antiseptic cleanser, was found to have greater efficacy in reducing the burden of a simulated water-based aerosol contaminant from the coats of working canines compared to towels saturated with water or dilute 2% chlorhexidine gluconate scrub.

Working canines can be tasked to environments rich in pathogenic microbiota. Following hurricanes [13,14] and floods [15], high levels of coliforms owing to raw sewage and wastewater system failures have frequently been detected in floodwater. Aerosolization of floodwater can occur both naturally and during boat operations. Additionally, urban environments are frequently contaminated with pathogenic microorganisms [16]. Air quality in urban centers is closely linked to local water quality; while diverse in microbiota, urban aerosols frequently carry pathogenic bacteria and viruses associated with sewage and wastewater treatment [17]. Working canines contaminated with aerosolized contaminants may accidentally ingest pathogenic microorganisms through self-grooming behaviors leading to gastrointestinal disease. Cross-contamination of human personnel with pathogenic microorganisms from a working canine’s exterior coat (fomite transmission) may place these individuals at risk for infection as well. Working canine decontamination is therefore essential to protecting both canine and human health.

Decontamination terminology is inconsistently utilized in medical literature [18]. Lack of standardized terminology can introduce confusion and result in poor compliance with recommendations. Although the term “decontamination” has been used as a collective phrase, the US Centers for Disease Control and Prevention [19] provide specific definitions for components of the decontamination process including sterilization (destruction or elimination of microbial life carried out in a health-care facility); disinfection (elimination of pathogens on inanimate objects); and cleaning (removal of visible soilage) [19]. Therefore, a decontamination strategy may incorporate both removal and/or inactivation of a pathogen. The study assesses the potential for physical removal of water-based aerosol particulates using a wipe-based procedure.

The decontamination potential of wipe-down procedures has been examined in review previously [20]. Suggested mechanisms for wipe-down include the use of a pre-soaked towelette, such as the one utilized here, as it was found to achieve greater physical removal of contaminating substances as compared to other procedures. Authors reported several factors impacting efficacy of the wipe-down procedure including pressure applied, nature of towelette, ratio of disinfectant to towelette, and others. Clearly the physical structure of the towelette will impact the amount of cleanser contained and subsequently released onto the surface of the dog’s coat. Although the current study was designed to investigate physical removal of contaminants, future studies should evaluate differences in wipes related to structure and disinfectant holding capacity to determine the maximum burden that a saturated towelette can retain prior to redistribution onto the dog’s coat. Prior work has demonstrated that the wiping process may dislodge contaminants and spread it over a larger area [21]. Thus, the impact of the biocidal capacity of the cleanser must be also carefully evaluated. The ideal wipe-down procedure should employ towelettes that can achieve physical reduction while utilizing a cleanser with appropriate biocidal action for the remaining contaminants. The selected cleansers should be assessed for both speed and spectrum of microbiological impact, as suggested by Sattar and Maillard (2013).

Decontamination is essential to prevent or limit direct and secondary exposure to toxins and pathogens encountered during field operations and is often performed multiple times a day [3]. Serial decontamination, while necessary, can disrupt the working canine’s epidermal barrier and diminish the protective effects of healthy skin and coat [3] increasing the likelihood of absorption of hazardous materials through the skin. Human studies demonstrate that repeated use of soap damages protein and lipids in the skin’s stratum corneum, leading to detectable dryness, redness, and irritation and increased skin permeability [22,23]. Prior canine studies [24,25] identified impacts of washing on skin by measuring dermal pH and barrier function of the epidermis as measured by trans epidermal water loss (TEWL; Discepolo) and the Canine Atopic Dermatitis Extent and Severity Index (CASESI; Zoran). Discepolo et al. demonstrated that cleanser selection can have a significant impact on the skin barrier with a single use, with Dawn^®^ dish soap causing more significant dermal effects than Nolvasan or Betadine [24]. Zoran demonstrated that repeated washing resulted in mild to moderate skin irritation in dogs in as little as 4.9 days using Dawn, which contains sodium lauryl sulfate (SLS), a high-anionic surfactant known to cause skin disruption in humans [26]. Interestingly, we found that use of wipe-down procedures employing disposable towels saturated with water, 2% CHX, or 7.5% povidone-iodine solution had no adverse impact on dermal pH. Decontamination using such field-expedient procedures can balance the need for frequent decontamination with preservation of canine skin integrity, with more traditional detergent-based decontamination reserved for the end of a work cycle.

The working canine’s coat provides a natural barrier to contamination, reducing direct skin exposure to some contaminants. Decontamination procedures that utilize large amounts of water have the potential to cause a “wash-in” effect whereby the decontamination procedure itself or the cleaner/biocidal enhances the penetration of contamination through the hair or into the skin [19,20]. An effective wipe-down procedure reduces the burden of aerosolized contaminants on the working canine’s exterior cut, limits the amount of clean water needed for field decontamination when resources are limited, and may even minimize the “wash-in” effect when performed frequently prior to traditional, water-intensive decontamination methods at the end of a work cycle.

While previous work has shown that working canines are readily exposed to and contaminated with oil-based agents through direct contact [10,11], we are only just starting to understand the risk of contamination with water-based aerosols present in the environment. A simple field-expedient wipe-down procedure utilizing disposable towels saturated with 7.5% povidone-iodine solution may effectively reduce the burden of water-based aerosol contaminant on the coats of working canines without adversely affecting dermal pH. Further work is needed to define appropriate exposure thresholds for performing wipe-down vs. traditional decontamination during working canine field operations.

The current study does include some limitations. Variability in coat type and structure across dog breeds may impact broad application of these findings. Future work should include representative populations across common dog breeds, not just those typically used in working disciplines. Additionally, the canines utilized were two facilities with similar management, diet, and housing conditions. Factors impacting coat quality may result in variation for effective contaminant removal. It is possible that oil content, sebum level, and other factors commonly known to impact dog coats may interfere with or improve contaminant removal due to changes in hydrophobicity. More work is needed to further investigate these areas and develop standards for best practices. Despite these limitations, the present study provides a novel data set for physical contaminant removal of water-based particulates on the coats of common working dog breeds.

## 5. Conclusions

These novel data contribute key findings to our current understanding of canine decontamination procedures and methods. Current public health concerns have highlighted an increased need for validated techniques to properly care for canines who may be exposed to aerosolized particulates. Working dogs, companion dogs, and the veterinary staff who may be called upon for their medical care need improved understanding for techniques to remove contamination of the coat. A commonly used and widely available veterinary cleanser, povidone-iodine scrub, effectively removed aerosolized particulates from coats of common dog breeds. Future investigations should identify impacts to skin and coat health from repeated (daily) use of these cleansers as well as biocidal efficacy of povidone-iodine scrub again bacterial and viral pathogens.

## Figures and Tables

**Figure 1 animals-11-00120-f001:**
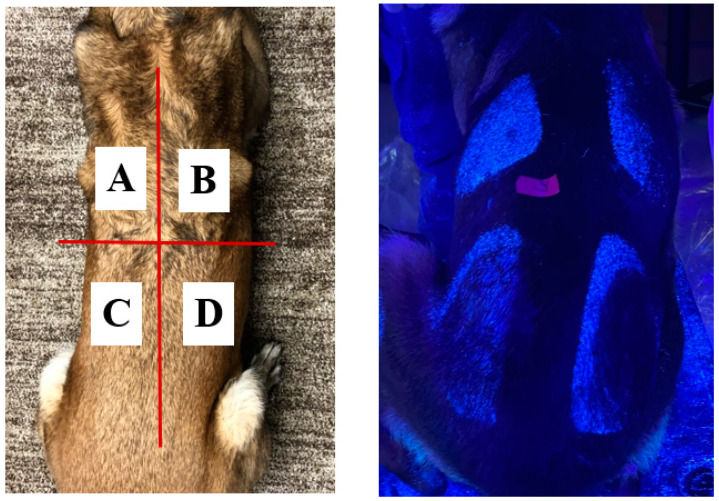
Canine dorsal area divided into quadrants for testing of a wipe-down procedure by removal of a simulated aerosolized contaminant. A = score 0 (unwiped control); B = Score 2; C = Score 1; D = Score 3.

**Figure 2 animals-11-00120-f002:**
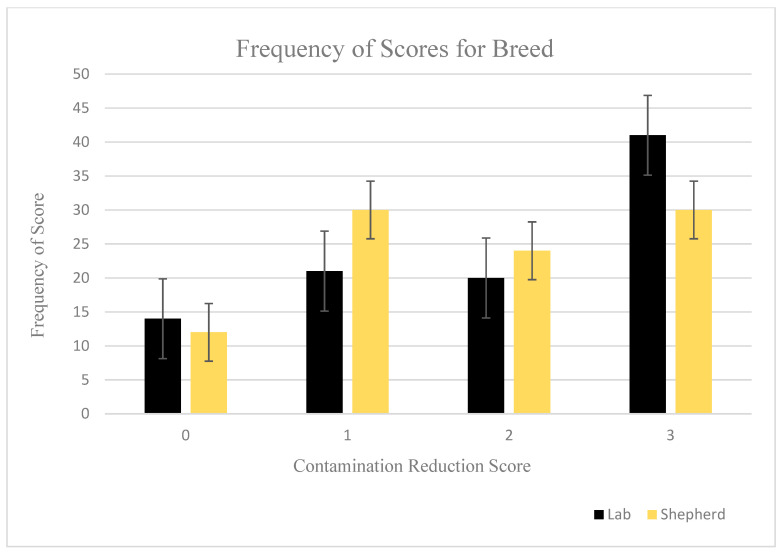
Reduction scores ^1^ using a simulated aerosolized contaminant were unaffected by breed (*p* = 0.6417). ^1^ Contamination reduction scores assigned as follows: 0 ≤ 25% contamination reduction; 1 = 25–50% contamination reduction; 2 = 51–75% contamination reduction; 3 ≥ 75% contamination reduction.

**Figure 3 animals-11-00120-f003:**
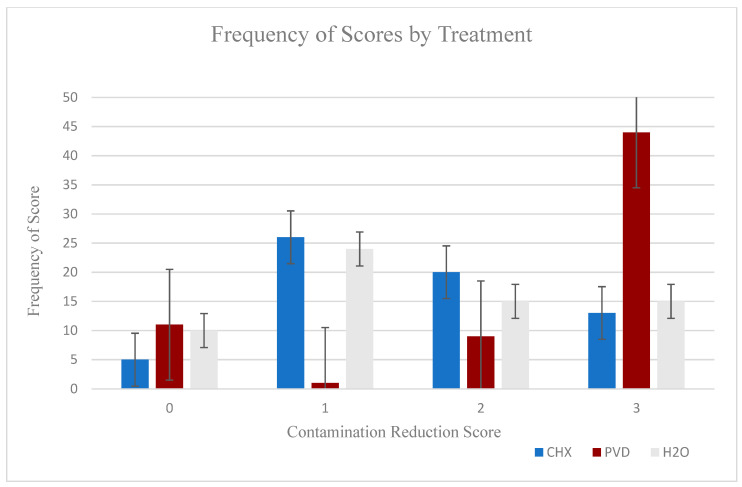
Cleanser selection impacts reduction scores ^1^ using a simulated aerosolized contaminant (*p* < 0.0001). ^1^ Contamination reduction scores assigned as follows: 0 ≤ 25% contamination reduction; 1 = 25–50% contamination reduction; 2 = 51–75% contamination reduction; 3 ≥ 75% contamination reduction.

**Table 1 animals-11-00120-t001:** Frequency of successful ^1^ wipe down utilizing common working breeds and veterinary antiseptic cleansers.

Breed	Water	Povidone-Iodine	Chlorhexidine Gluconate	*p* Value
**Labrador retriever**	7 ^a^	26 ^b^	8 ^a^	<0.001
**German shepherd**	6 ^a^	21 ^b^	3 ^a^	<0.001

^1^ Success = contaminant reduction score of 3. ^a,b^ Unlike superscripts indicate statistical significance.

## Data Availability

Data from this study are available upon request from the corresponding author. The data are not publicly available due to privacy concerns.

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
