# Peer review of "Removal of Aerosolized Contaminants from Working Canines via a Field Wipe-Down Procedure"

_animals, 2021, doi:10.3390/ani11010120_

Round 1
Reviewer 1 Report
There is too little scientifically needed information in this manuscript, which it's sloppy.
First of all, the reviewer couldn't find any figures in the text.
Other issues are listed below.
1) The authors must exemplify the unconventional harm caused by contamination.
2) It should be noted that the two chemicals, 2% chlorhexidine gluconate scrub (CHX) and 7.5% povidone-iodine 89 scrub (PVD), are completely harmless to dogs.
3) The reviewer doesn't understand the meaning of measuring pH. Is the dermal pH good between 8 and 9?
4) The effect of wiping with water is unknown.
Author Response
There is too little scientifically needed information in this manuscript, which it's sloppy.
Thank you for the opportunity to address your revisions. We have carefully examined each of your comments and amended the manuscript accordingly where appropriate. We appreciate your time and consideration.
First of all, the reviewer couldn't find any figures in the text.
Thank you for the opportunity to address this concern. It appears that the files containing the figures were not available for your review due to some unknown error in the system. The submission process requires the figures to be uploaded in a separate file. I apologize that they were not viewable. I have added them at the end of the manuscript for ease of your review. Please see Figures starting at line 344.
Other issues are listed below.
- The authors must exemplify the unconventional harm caused by contamination.
Thank you for the opportunity to address this concern. We have added additional text with supporting citations to clarify the types of hazards and health risks associated with contamination. Please see lines 44-47 and lines 51-53.
- It should be noted that the two chemicals, 2% chlorhexidine gluconate scrub (CHX) and 7.5% povidone-iodine 89 scrub (PVD), are completely harmless to dogs.
Thank you for the opportunity to address this concern. CHX and PVD are relatively benign products when used at appropriate concentrations. However, it would be factually incorrect to state that CHX and PVD “are completely harmless to dogs” as both have been shown to induce skin pathology at high concentrations. We have added additional information on PVD and CHX in Lines 65-74 with appropriate supporting citations.
- The reviewer doesn't understand the meaning of measuring pH. Is the dermal pH good between 8 and 9?
Thank you for the opportunity to clarify this. Measuring dermal pH has long been associated with barrier function and health of skin in humans and other mammals. Canines skin has been demonstrated as basic (Rippke et al., 2002; Matousek, 2003) and has been measured >7 on the pH scale. However, it is important to note, that like in humans, anatomical location, age, diet, stress level, cleansers and gender are theorized to have an effect on dermal pH. To measure pH in this study we used a flat tipped hand-held pH meter (manufacturer information found in materials and methods) that was designed for dermatological use. Canines experienced similar scenarios during testing and pH was taken at the same anatomical location on all canines. As changing dermal pH can impact the coat, our purpose for measuring dermal pH was to identify any impact off varying pH on efficacy of the wiping procedure. No effect of pH was identified.
- The effect of wiping with water is unknown.
Thank you for the opportunity to clarify. The water was utilized as a control to determine if the CHX or PVD solutions achieved greater physical reduction. A dry towelette would not have been appropriate as a control due to differences in load capacity. Biocidal studies are underway to evaluate the impact on pathogen load when utilizing the two cleansers. Additional language has been added at lines 173-196 to provide greater context and explanation for the benefits of the wipe-down procedure as well as impacts of saturation. The second wipe (the rinse step) was incorporated to remove any soapy residue from the coats of the dogs and to mimic real-world recommendations for use.
Reviewer 2 Report
Highlights/simple Summary: it should be revised splitting simple summary and highlights and deleting numbers among sentences.
Materials and Methods: explain why authors chose Labrador and German Shepherd and how they established the number of animals involved in the study
I suggest the authors follow ARRIVE Guidelines to improve study report.
The solution used to simulate the contamination and the protocol adopted were previously validated? What kind of contamination are the authors acting out? How could the removal of droplet fluorescence be indicative of the efficacy of a decontamination method without a validation with bacterial or chemical substances even only on inanimate surfaces?
After "Discussion" a paragraph with the limitations of the study would be appreciated.
Author Response
Highlights/simple Summary: it should be revised splitting simple summary and highlights and deleting numbers among sentences.
Thank you for the opportunity to address your concerns. It appears that an editing issue merged two files which caused the confusing mix. We have corrected that issue.
Materials and Methods: explain why authors chose Labrador and German Shepherd and how they established the number of animals involved in the study
Thank you for the opportunity to address your concerns. We have revised the language of the Materials & Methods for a better explanation. Please see line 87-89. As we felt it important to use working dogs for this study, the number utilized was a convenience sample with the maximum number of dogs available for our use. Additionally, the number is larger than prior canine decontamination studies available in the literature (Powell et al, 2018; Venable et al., 2017).
I suggest the authors follow ARRIVE Guidelines to improve study report.
Thank you for the suggestion. We have completed the ARRIVE Guidelines checklist (see attached) with line numbers for your convenience during the review.
The solution used to simulate the contamination and the protocol adopted were previously validated? What kind of contamination are the authors acting out? How could the removal of droplet fluorescence be indicative of the efficacy of a decontamination method without a validation with bacterial or chemical substances even only on inanimate surfaces?
Thank you for the opportunity to address your concerns. We have added additional language in the Discussion section to further explore the potential efficacy of using wipes (towelettes). Please see lines 175-195. We have also added language clarifying the assessment of droplet removal in the Materials & Methods section. Please see lines 120-122. In addition, we have described the need for biocidal impact studies to investigate properties associated with pathogen destruction, as correlated to decontamination per the CDC. Please see lines 196-200.
After "Discussion" a paragraph with the limitations of the study would be appreciated.
Thank you for the opportunity to address your concern. We have included the suggested paragraph. Please see lines 236-245.
Reviewer 3 Report
It is a nice study with relevant data - but I found this during my research - please check this with the editors https://www.preprints.org/manuscript/202011.0229/v1
Author Response
It is a nice study with relevant data - but I found this during my research - please check this with the editors https://www.preprints.org/manuscript/202011.0229/v1
Thank you for the opportunity to address your concern. The preprint option is offered through the publisher and is available as part of the review process for this journal.
Round 2
Reviewer 2 Report
Thank you to the authors for their answers and explanations. I think the paper can be published in the present form.